# A Cost-Effective Nucleic Acid Detection System Using a Portable Microscopic Device

**DOI:** 10.3390/mi13060869

**Published:** 2022-05-31

**Authors:** Chengzhuang Yu, Shanshan Li, Chunyang Wei, Shijie Dai, Xinyi Liang, Junwei Li

**Affiliations:** 1Hebei Key Laboratory of Smart Sensing and Human-Robot Interactions, School of Mechanical Engineering, Hebei University of Technology, Tianjin 300130, China; 201811201004@stu.hebut.edu.cn (C.Y.); 201811201008@stu.hebut.edu.cn (C.W.); 2State Key Laboratory of Reliability and Intelligence of Electrical Equipment, Hebei University of Technology, Tianjin 300130, China; 3Institute of Biophysics, School of Health Sciences and Biomedical Engineering, Hebei University of Technology, Tianjin 300130, China; 202121103006@stu.hebut.edu.cn

**Keywords:** portable, fluorescence, microscope, meat products authenticating

## Abstract

A fluorescence microscope is one of the most important tools for biomedical research and laboratory diagnosis. However, its high cost and bulky size hinder the application of laboratory microscopes in space-limited and low-resource applications. Here, in this work, we proposed a portable and cost-effective fluorescence microscope. Assembled from a set of 3D print components and a webcam, it consists of a three-degree-of-freedom sliding platform and a microscopic imaging system. The microscope is capable of bright-field and fluorescence imaging with micron-level resolution. The resolution and field of view of the microscope were evaluated. Compared with a laboratory-grade inverted fluorescence microscope, the portable microscope shows satisfactory performance, both in the bright-field and fluorescence mode. From the configurations of local resources, the microscope costs around USD 100 to assemble. To demonstrate the capability of the portable fluorescence microscope, we proposed a quantitative polymerase chain reaction experiment for meat product authenticating applications. The portable and low-cost microscope platform demonstrates the benefits in space-constrained environments and shows high potential in telemedicine, point-of-care testing, and more.

## 1. Introduction

With the rapid development of nucleic acid detection technology, polymerase chain reaction (PCR), invented by Mullis in 1983 [1], has been widely employed. In brief, an optical (fluorescence or UV) detection system is required to read the data from real-time fluorescence intensity of a centrifuge-tube [2], emulsion droplets [3], or microfluidic chips [4]. Among these detection devices, the portable fluorescence microscope has recently become an indispensable instrument for on-site nucleic acid detections, especially during the outbreak of COVID-19. Beyond nucleic acid detection, fluorescence microscopy is also an important tool in biomedical research for the visual analysis and study of molecules, cells and tissues [5], and microfluidic applications [6]. The invention of the microscope in the 16th century led to the rapid development of biology, medicine and other disciplines. In the 1910s, the invention of the fluorescence microscope further accelerated the development of microscopic subjects at a faster-than-ever pace. With the rapid advancement of technology, many high-end powerful microscopes have been developed, such as the stereoscopic microscope [7], the inverted fluorescence microscope [8], the scanning electron microscope (SEM) [9], laser scanning confocal microscopy (LSCM) [10] and the atomic force microscope (AFM) [11], etc. A commercial fluorescence microscope consists of a base with a precision optical path, a wide range of mechanical moving stages, a set of precision objectives with switchable magnification, a fluorescence light source and a charge-coupled device (CCD) camera. In addition, a set of filters and an ultra-high pressure mercury lamp (UHP) are also needed when observing fluorescence. These configurations allow for versatile, high-quality imaging in a stable environment, but also yield microscope systems that are bulky and costly. The lack of portability limits the use of microscopes in point-of-care diagnostics, and their cost limits their availability in resource-limited areas. Therefore, the development of portable and low-cost fluorescence microscopy systems is warranted.

For portability and cost-efficiency purposes, smartphone-based microscopes provide a potential straightforward solution for a broad range of applications. Several mobile phone-based portable microscopic imaging systems have been developed for point-of-care diagnostics [12]. For example, a mobile phone-based flow cytometer was developed to measure white blood cell density in human blood samples [13] and also to detect and count *Giardia lamblia* cysts by incorporating machine learning [14]. Another study demonstrates a handheld smartphone fluorescence microscope without additional optical filters [15] by designing a dual-functional polymer lens that performs both optical imaging and filtering. Using a single lens and a built-in application, the Pocket MUSE [16] and our previous work [17] both demonstrate the compatibility, portability, and user-friendly sample preparation strategies for various applications. Beyond the academic literatures, there are also some commercially available handheld microscopes such as ioLight and Dino-lite microscopes. The ioLight microscope has a fluorescence module, which is mainly used for the qualitative classification of bio-samples. With a focus on electronic circuit inspections rather than bio-related samples, Dino-lite microscopes are mostly designed for bright-field applications.

Generally speaking, the way to achieve a low-cost and portable fluorescent microscope system is to replace expensive components with inexpensive alternatives to achieve similar functionality. Current developments in consumer-oriented manufacturing technologies, such as 3D printing, are facilitating individual researchers to customize laboratory equipment or build entire devices from scratch [18]. In parallel, ultra-low cost light-emitting diodes (LEDs) are now available that provide enough power to excite fluorescent material for optical imaging [19,20]. Similarly, with the rapid development of optics and manufacturing, high performance image sensors such as complementary metal oxide semiconductor (CMOS) image sensors and optical components such as lenses and spectral filters are available at a low cost [21]. Several compact image sensors have thus become available [22,23]. For example, mini-microscope systems built on polymethyl methacrylate (PMMA) architecture allow real-time monitoring of cell behavior in a thermostat [24]. Lens-free computational microscopes [25] have also proven to be suitable for low-cost portable architectures, using the lens-free imaging system for real-time imaging analysis of human malaria parasites [26].Based on our previous work on bright-field portable microscope prototypes [17,27] on the applications of dairy somatic cell counting and microalgae cell counting, here we have further developed a high-resolution fluorescence microscope system that is low-cost, portable and capable of adjustable mechanical movements. Compared with previous work, the proposed prototype has a compact XYZ stage to facilitate the operator to perform optical investigations in a larger area. Regarding cost efficiency, our fluorescence microscopy system consists of an assembled mechanical moving stage (costing approximately USD 30/set), a webcam-modified imaging unit (costing approximately USD 50/set), a battery-operated LED light source (costing < USD 1/set) and a set of interchangeable filters (costing < USD 5/set), which means the entire device costs no more than USD 100.

To evaluate the capability of the portable fluorescence microscope, we proposed a quantitative polymerase chain reaction experiment for meat product authenticating, using a microfluidic chip. In recent years, researchers have also made efforts in on-chip PCR technology. Continuous microfluidics-based on-chip PCR systems were developed for the detection of Epstein–Barr virus [28] and multiplex amplification of target genes of periodontal pathogens [29]. Self-discretization chips [30] and multi-color fluorescence chips [31] based on digital PCR have also been developed for the quantitative detection of nucleic acids. In addition, fast temperature response [32] and compact construction [33] are also important in point-of-care diagnostic devices. Herein we present detailed instructions for the assembly of a portable microscopic device. Additionally, the method to determine the field of view and magnification was provided by hardware/software configurations. The fluorescence images from splitting channels (blue, green) were shown to evaluate the performance of fluorescent microscope. Finally, a case study in nucleic acid detection using a commercial Multienzyme Isothermal Rapid Amplification (MIRA) assay kit was demonstrated in this work.

## 2. Materials and Methods

### 2.1. Designing and Operation of the Microscope

Our goal was to design a portable miniature fluorescence microscope. It had to provide sufficient precision to focus on microscopic objects while being mechanically stable. To control costs, 3D printed parts were used for assembly to make it affordable and deployable in low-resource situations. Importantly, it also had to provide fine resolution and image quality for most applications to enable bright-field and fluorescence channel measurements.

The overall structure of the portable microscope is shown in Figure 1A. It has a compact dimension (~10 cm × 12 cm × 16 cm) and is highly portable (~355 g), with a cost of around USD 100. The body of the portable microscope is assembled from 3D printed parts, including a base, a manual rotor, and a set of parts for the sliding stage. For lighting, a battery-operated white LED bead (Geesled, Shenzhen, China) is mounted on top as the bright-field light source and another colored LED bead (selected according to the excitation wavelength of the fluorescent substance) on the side as the excitation light for fluorescence observation. Coin cell batteries (3V, CR2032, Nanfu, Nanping, China) are placed in the battery box with an on/off switch for easy control of the light source.

The base structure of the microscope was fabricated by 3D printing using white photosensitive resin with a layer thickness of 200 μm. The model was designed in the 3D drawing software SolidWorks 2016 (Dassault Systems, Paris, France). All holes and threads for bolt and screw assembly can be printed out directly without additional machining. An *X*-*Y* axis sliding stage was assembled with high-precision miniature slides and springs. The movement of the *Z*-axis depended on the manual rotor that fits into the base.

A commercially available webcam (640 × 480 pixels, KS2A103, Kingcent, Shenzhen, China) was mounted on the bottom. The bracket extending from the bottom was used to install the LED light source. For the webcam to perform the functions of a microscope, the lens was inverted to achieve magnification rather than the magnification mechanism employed by the camera. Images were obtained from a laptop with the webcam connected through a universal serial bus (USB) port, using a webcam program.

To obtain clear microscopic images from targets with different sizes, a sliding stage was assembled based on a combination of 3D-printed components and four micro-slides (MGN7C, Fangbo Automation, Wenzhou, China). Among the three freedom-of-degrees, the *XY* direction is designed to adjust the focusing area, while the *Z* direction is designed for focusing and magnifications. The demonstration video of the operations for the *XYZ* movement is provided in the Appendix A.

As shown in Figure 1B, a set of two slides were assembled with the 3D-printed parts for movement in the *X*-direction and another set for movement in the *Y*-direction. In order to limit the volume and to have a large adjustment range, each slide was machined into being 50 mm long to allow a movement range of approximately 2 cm × 2 cm in area after assembly. Two hand-screwed bolts (M5 × 25) were used for adjustment and four springs for bringing the stage into position. Note that the bracket should be designed to be dispersed and assembled, otherwise it will not be assembled due to interference. For *Z*-direction adjustment, utilizing the bolt and nut mechanism, a 3D-printed nut fits into the threads on the base, and the stage can be focused with the rotation of the manual rotor (Figure 1C). In addition, to prevent the stage from rotating, when the manual rotor turns, the part of the sliding stage that fits into the base was specifically designed as an irregular column. The assembled sliding stage can be loaded and unloaded as a single unit.

The microscopic imaging unit of our portable microscope is converted from a commercial webcam. Figure 1C illustrates the hardware part and the modification process of the camera module of the portable microscope. The lens was flipped over and secured to the CMOS by a section of silicone tubing. The modified webcam was fixed to the base by bolts and nuts to ensure stable imaging. The camera was connected via a universal serial bus (USB) port and the images were obtained from a laptop using a custom camera program. The camera program is able to save images and image sequences in JPEG format and video data in an audio video interleaved (AVI) format.

Figure 1D illustrates the operation of the portable microscope. The sample was placed on the stage, and the approximate observation area was aligned with the lens. The manual rotor was rotated clockwise or counterclockwise to focus the sample. The movement of the sample in the X-Y plane can be adjusted by turning two hand-screwed bolts.

To evaluate the essential imaging performances (field of view, and spatial resolution) of the portable microscope, a commercial hemocytometer (XB-K-25, Shanghai, China) was used for calibration. Polystyrene microspheres of different diameters (5, 10, 20, 50 μm in diameter, Baseline, Tianjin, China) were used to evaluate the spatial resolution of the portable microscope. Bright-field images were captured, and pixel intensities were measured with ImageJ software (National Institutes of Health, Bethesda, MD, USA). The field of view was calculated by imaging the grids on the hemocytometer.

Figure 1E illustrates the system configurations of our portable nuclei acid detection system, which includes a heating unit (a heater, a temperature sensor, and a microcontroller), a fluorescence microscopy, and a microfluidic chip. Figure 1F shows the typical temperature profiles for nuclei acid amplifications. The real time temperature shown in Figure 1F could be controlled by the microcontroller in Figure 1E.

### 2.2. Construction of Meat Products Authenticating Experiment

#### 2.2.1. Fabrication of Reactors

To facilitate observation, sealable microfluidic chips were fabricated to replace the centrifuge tubes used in commercial PCR instruments. First, polydimethylsiloxane (PDMS, Dow Corning Sylgard 184, Dow Corning Corporation, Midland, MI, USA) was poured onto a clean slide. Note that the slides need to be silanized to strengthen the hydrophobicity for PDMS peeling. After curing at 80 °C for 2 h, the PDMS was peeled off the slides. Next, according to the experimental requirements, one or several holes of 3 mm diameter were punched. Then, the PDMS chip was bonded on top of the glass substrate by standard plasma bonding. Finally, after adding the reaction solution, an adhesive sealing film was placed on the chip to seal it. Furthermore, a miniature thermocouple (24 V, 110 °C, Yidu Electronics, Shanghai, China) connected with a microcontroller was placed on the side of the glass slider to satisfy the temperature requirements of nucleic acid amplifications shown on Figure 1F.

#### 2.2.2. Preparation of Meat Samples and DNA Extraction

The meat used in this study was duck purchased from a local supermarket. DNA was extracted from the tissues and purified using MightyPrep™ (Takara, Dalian, China) according to the manufacturer’s protocols. First, 25 mg of chopped test samples were placed in a 1.5 mL tube. Next, 100 μL of sterilized water was added, capped tightly, and processed at 95 °C for 5 min. Finally, centrifugation (12,000 rpm, 5 min) was performed, and the supernatant was taken as the test sample. A reagent blank was used as a negative control for each DNA extraction. The concentration and purity of the extracted DNA were quantified using an ultraviolet (UV) spectrophotometer (UV5-Nano, Mettler Toledo, Zurich, Switzerland). All DNA samples were stored at −20 °C before use.

#### 2.2.3. PCR Amplifications

A duck-derived commercial thermostatic assay kit (Amp-Future, Weifang, China) was used in this work. Following the instruction guides, the primers were mixed with the solution containing the recombinase, polymerase, and the buffers, following which 2.5 μL of sample DNA was added; the preparation (total volume 50 μL) was then kept at 39 °C for 20 min. The acquisition of fluorescent images was carried out every 30 s.

## 3. Results and Discussions

### 3.1. Evaluations of FOV and Magnification

Essentially, a larger field of view is achieved at the expense of magnification. Therefore, it was important to make a balance according to the requirements of real applications. Figure 2(Ai) shows the microscopic images of a commercial hemocytometer captured by a Nikon microscope. Here, in Figure 2(Bi), the microscopic image was captured under the condition of the maximum field of view and the minimum magnification. Based on the design conception of the hemocytometer, the size of the minimum grid and the minimum double line gap is 50 μm, and 10 μm, respectively. Thus, the field of view in Figure 2A is estimated as 840 μm × 630 μm, namely 0.5292 mm^2^ (Appendix A). In the same condition, the minimum magnification of our portable microscope device is estimated at 164×. The total display magnification for the microscopy can be expressed by [34]:(1)$$M=\frac{N_{m}}{N_{s}}\cdot R_{p},\;$$
where M is the total display magnification, $N_{m}$ is the number of pixels for the monitor image width, $N_{s}$ is the number of pixels for the sensor image width, and $R_{p}$ is the pixel size ratio. The pixel size ratio $R_{p}$ is defined as:(2)$$R_{p}=\frac{S_{\mathrm{mon}}}{S_{\mathrm{sen}}},\;$$
where $S_{\mathrm{mon}}$ is the pixel size of the monitor and $S_{\mathrm{sen}}$ is the pixel size of the CMOS sensor.

Figure 2(Bii–Bv) demonstrates the microscope images of commercial polystyrene particles captured by our device (M12 adaptor, focal length 3.8 mm). Here, the particle diameters are 5 µm, 10 µm, 20 µm and 50 µm, respectively. As a comparation, the microscopic images of hemocytometer and polystyrene particles from commercial microscope (20× objective lens) were investigated, as shown in Figure 2(Aii–Av). Moreover, the FOV and magnification could be further improved by adjusting the focal length or work distances of the microscopic system. Basically, it is certain that, the portable microscope device is able to provide satisfied images with on-demand FOV and magnifications. Thus, the low-cost and portable microscopic device could be comparable to commercial microscopes for many applications.

To evaluate the spatial resolution of the microscope, the light intensity of microscopic images in Figure 2 were analyzed, as shown in Figure 3. Normalization was carried out to ensure that the intensity value ranged between zero and one. In the plot of normalized light intensity as a function of position, we provide plots from 5 µm, 10 µm, 20 µm, and 50 µm particles. It can be seen that the pulse width increases with particle size. From Figure 3A,B, it can be seen that the pulse width which represents the particle size is very close to each other. Even for the 5 µm particles we could obtain a clear microscopic image from the portable microscopic device. Thus, the spatial resolution of our device could be as small as 5 µm scale. We also performed imaging observations on real applications, as shown in Figure 3C,D. The microscopic images of human hepatoma cells (HepG2, 25~30 µm in diameter, Easy Biotech, Shanghai, China) can be clearly observed in Figure 3C. The images of microalgal cells (Health Biotech, Nanjing, China) with a diameter of approximately 3 µm are shown in Figure 3D. Although the spatial resolution of our device does not reach 3 µm, we can still observe the dynamics of microalgae.

### 3.2. Fluorescence Imaging

One critical function of a low-cost and portable microscope is its fluorescence capability. A fluorescent substance absorbs excitation light into the excited state and then emits light with a wavelength longer than the excitation light. To observe fluorescence images, the commercial microscopes are usually equipped with an ultra-high-pressure mercury lamp (UHP) and a dichroic mirror. The UHP is capable of emitting high-intensity ultraviolet and blue-violet light, which is sufficient to excite various fluorescent targets. The dichroic mirror could filter the excitation light to ensure that the emitted light can be received for imaging. These components ensure the imaging quality of the commercial microscope. However, with a bulky size, the cost of a commercial microscope is generally no less than several hundred dollars, making them unsuitable for low-cost and portable applications.

LEDs have a very narrow band spectrum (20–40 nm), so it is possible to replace UHP as the light source to omit the dichroic mirror. Instead, emission filters could be used to improve the specificity of the optical spectrum. Therefore, we used RGB LED beads as the excitation light source and low-cost filters were fixed on the lens after easy assembly. Figure 4A shows the working status and optical principles of our fluorescence imaging device. To prevent the excitation light from bleed-through into the camera path, the light source was positioned at 45°, relative to the plane of the objective in this work.

As shown in Figure 4B,C, green fluorescent microspheres (ex/em 460/515 nm, 20 μm) and red fluorescent microspheres (ex/em 554/610 nm, 20 μm) were used as a target to show the performance of fluorescence imaging. Take Figure 4B for example, a 465 nm blue LED and a 510 nm filter were used as the light source to excite the green, fluorescent microspheres. With a 510 nm filter, the blue light was not well blocked, thus, a blue background is observed. Anyway, the blue background did not affect the post-processing of fluorescent images at all. Since each RGB image has three channels, we could obtain the green channel component of the original fluorescent images obtained from our portable microscopic device, as shown in the third subfigure of Figure 4B. In the same way, we obtained the microscopic images of red fluorescent microspheres under bright-field, the 525 nm LED fluorescent field, and the red channel component of the fluorescent image, as shown in Figure 4C.

To analyze the fluorescent images in Figure 4B, we provided the fluorescent intensity plot as a function of position, as shown in Figure 4D. It can be seen that, the relative intensity between the green, fluorescent microspheres and the blue background is enlarged by RGB channel splitting. In the same way, Figure 4E shows the fluorescent intensity of red fluorescence particles compared with green background, before and after RGB channel splitting. Thus, the proposed RGB channel splitting operation can be well applied to analyze the fluorescence images obtained from our portable microscopic device.

### 3.3. Application of Meat Products Authenticating

We tested the sensitivity of the device and method by detecting serial dilutions of duck-derived DNA. As shown in Figure 5A, we tested six concentrations of samples, 10^5^ copies/μL, 10^4^ copies/μL, 10^3^ copies/μL, 10^2^ copies/μL, 10^1^ copies/μL, and blank control, where amplification curves were observed for concentrations 10^2^ copies/μL and above, indicating that the detection limits of our device and method were 10^2^ copies/μL.

We extracted DNA samples from duck products purchased from local supermarkets. To ensure the reliability of the experimental results, the experiment was set up with two identically configured experimental groups (test 1 and test 2), a positive control group (positive sample DNA in kit) and a blank control group (ddH2O instead of DNA samples). The blue LED was used as the excitation light for the experiments because the probes in the kit were labeled with carboxy fluorescein (FAM, ex/em 492/518 nm). As shown in Figure 5B,C, the fluorescence images of the green channel in test 1 were separated according to the method in the previous subsection. Only the images at minutes 10, 12, 14 and 16 are shown here, as this is the exponential growth period of test 1 amplification, where significant changes in fluorescence intensity can be observed. Since the number of DNA grows slowly in the early stage, the fluorescence intensity is basically constant from zero to ten minutes. Additionally, after the exponential growth period, the amplification gradually reaches saturation, and the fluorescence intensity is no longer enhanced. As shown in Figure 5C, the fluorescence intensity of the acquired images was plotted versus time to obtain the amplification curve. As expected, both samples extracted from duck products exhibited amplification curve profiles to the positive template, demonstrating the utility of our device and method. Such capability of the portable microscope provides a unique opportunity to easily investigate the source of meat products outside the laboratory.

To make the device more convenient for resource limited situations, we developed an Android application for grayscale analysis of fluorescence intensities. As shown in Figure 5D, we could upload the fluorescence image and calculate the range of grayscale, and the average value of the images. Based on the average grayscale at a specific time, we could distinguish positive samples from negative ones.

### 3.4. Advantages, Limitations and Future Works

Published works demonstrate a smartphone-based portable microscope by combining a microscope with a smartphone using the camera on the smartphone [12,13,14,15]. Comparing these efforts, we note that most of them require manual adjustment of the field of view when working. For micron-scale targets, manual adjustment may not be very convenient for the operators. Our device was designed with a compact XYZ stage that allows micron-scale movement, making it easy for the operators to find the best area of interest for optical observations. With an ultra-compact mechanical configuration, the proposed microscopic prototype is able to provide bright/fluorescent images such as a benchtop microscope. On the other hand, the limitation also lies in its ultra-compact mechanical configuration. To make the prototype more compact, the movement of Z-axis cannot provide a remarkable change of amplifications. In the proposed design, we need to change a lens or lens adaptor to obtain a larger amplification.

Regarding the on-chip PCR heating system, we chose the contact heating method as we used the MIRA method, which is a thermostatic and rapid amplification method. By calibration, the temperature can be kept constant, and the heating system is simplified. In addition, the background noise in fluorescence mode reduces the sensitivity of fluorescence microscopy to some extent, and this problem can be solved by optimizing the optical path and using an artificial intelligence image processing algorithm. We will make efforts to improve the sensitivity in our future work.

## 4. Conclusions

Here, we have demonstrated that it is possible to build a portable (~10 cm × 12 cm × 16 cm, ~355 g) microscope with bright-field and fluorescence modes under a USD 100 budget. With a specific design of optical path, the low-cost and portable microscope has a magnification and field-of-view similar to a commercial microscope. With a spatial resolution of 5 μm, the proposed portable microscope shows satisfied performance for fluorescence imaging in many bio-applications such as cell imaging or nuclei acid detections. Here, we analyzed the meat authenticating tests using real time fluorescence nucleic acid detection method. With a channel splitting operation for RGB images obtained from our portable microscopic device, the cost-effective nucleic acid detection system could recognize any changes in fluoresce intensities during the nuclei acid amplification process. In the future, our portable microscopes will be ideal for downstream applications in combination with microfluidic devices, especially in conditions of limited space and resources. Additionally, we can also foresee applications in diagnostics or telemedicine.

## Figures and Tables

**Figure 1 micromachines-13-00869-f001:**
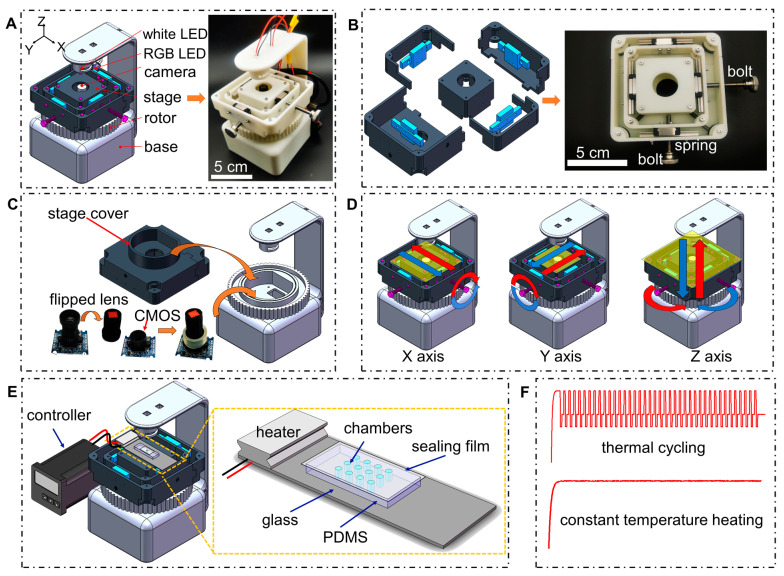
Design, assembly and operation of the portable nucleic acid detection system. (**A**) Three-dimensional illustration and physical prototype of the portable fluorescence microscope with LED light sources, web camera, sliding stage, manually rotor and a base. (**B**) Explosive view of the sliding stage assembled with the camera, together with the assembled sliding stage. (**C**) Schematic diagram of the microscopic system. (**D**) Demonstrations for the adjustment of sliding stage with three degree-of-freedoms. (**E**) Illustration of the system configurations including the heater, temperature controller, microscopic device and microfluidic chip. (**F**) Typical temperature profiles for nuclei acid amplifications.

**Figure 2 micromachines-13-00869-f002:**
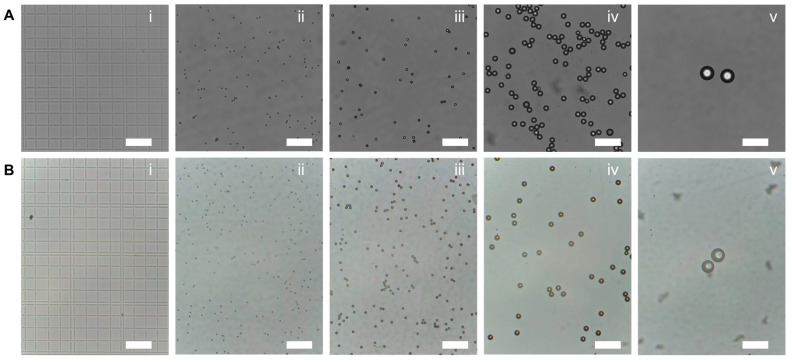
Performance of our portable microscopic device, with maximum FOV and minimum magnification, compared with commercial microscope. (**A**) Microscopic image of a commercial hemocytometer captured by Nikon microscope. (**B**) Microscopic image of a commercial hemocytometer captured by the portable microscopic device proposed in this work. From left to right, labeled with (**i**–**v**), stands for the microscopic image of a commercial hemocytometer, polystyrene particles with diameters of 5 µm, 10 µm, 20 µm and 50 µm, respectively. Scale bars are 100 µm.

**Figure 3 micromachines-13-00869-f003:**
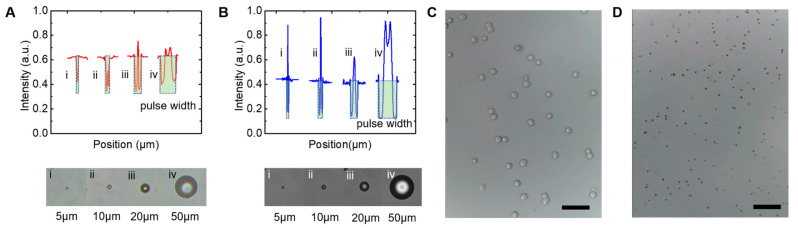
Plots of normalized light intensities as a function of microsphere diameter, for polystyrene particles with diameters of 5 µm, 10 µm, 20 µm, and 50 µm. (**A**) light intensities of the microscopic images from our device. (**B**) light intensities of the microscopic images from commercial microscope. (**C**,**D**) Image of human hepatoma cells (HepG2, 25~30 µm in diameter) and microalgae cells (around 3 µm in diameter) from our device, respectively. Scale bar are 100 µm.

**Figure 4 micromachines-13-00869-f004:**
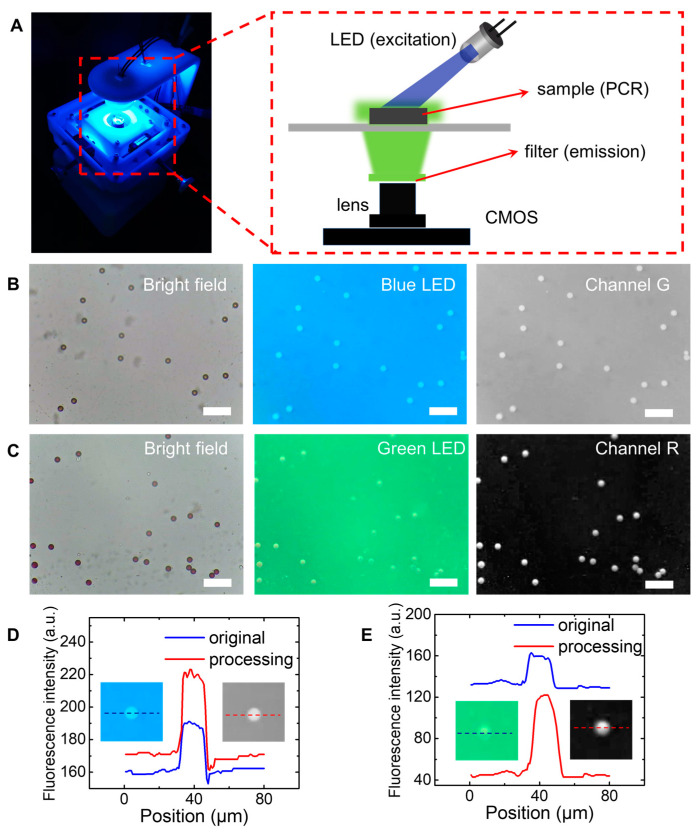
Performance of our portable microscopic device in fluorescence mode. (**A**) Working principles of fluorescence imaging of our device. (**B**) Microscopic images of green polystyrene particles using write LED and blue LED as light source. From left to right, are images from bright-field, blue LED field, and green channel. (**C**) Microscopic images of red polystyrene particles using write LED and green LED as light source. From left to right, are images from bright-field, green LED field, and red channel. (**D**) Fluorescence intensities from RGB images compared with green channel only. (**E**) Fluorescence intensities from RGB images compared with red channel only. Scale bars are 100 µm.

**Figure 5 micromachines-13-00869-f005:**
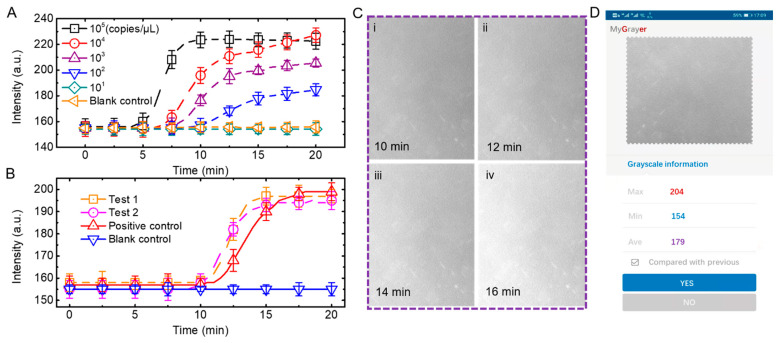
Application of duck products authenticating. (**A**) Results of real-time amplification of nucleic acids using concentrations of 10^5^ copies/μL, 10^4^ copies/μL, 10^3^ copies/μL, 10^2^ copies/μL, 10^1^ copies/μL and blank control, respectively. (**B**) Fluorescence intensity as a function of time during nuclei acid amplifications. (**C**) Images of the green channel (emission) after separation were captured at 10 min, 12 min, 14 min, and 16 min. (**D**) Android application for grayscale analysis of fluorescence intensities.

## Data Availability

The data that support the findings of this study are available from the corresponding author, upon reasonable request.

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
