# Peer review of "A Cost-Effective Nucleic Acid Detection System Using a Portable Microscopic Device"

_micromachines, 2022, doi:10.3390/mi13060869_

Round 1

Reviewer 1 Report

In this work, authors have presented a portable fluorescent microscopy system. This is an interesting topic and authors have provided an interesting manuscript. however, there are a few points that need to be addressed:

  1. In the Introduction, authors should provide background information on previous work on point of care or portable microscope. This also includes systems which use phones and built in apps to magnify like Pocket MUSE. These kind of apps and devices have similar application as is intended by this mini-microscope system.
  2. Authors need to provide how their work is different from previous work and how is it more cost effective.
  3. Please compare your device efficiency with other portable microscopes like iolight microscope; Dino Light microscopes
  4. What are the limitations of the device? 
  5. Authors should add error bars in Figure 5a.
  6. How does the microscope perform with real world samples like blood smear; etc?

The work is not novel and authors should provide the motivation for the work, especially when there are commercially available portable microscopes

Reviewer 2 Report

This manuscript entitled “A cost-effective nucleic acid detection system using a portable microscopic device” by Chengzhuang Yu et al.

This manuscript reports a portable microscopic system that can be used to observe fluorescent droplets for analyzing digital polymerase chain reaction (digital PCR). The followings are some raised questions. The authors may need to clarify these questions before this manuscript can be accepted. 

  1. The authors developed an on-chip PCR technology by employing a thermal heater. Recently, several efforts of on-chip PCR devices have been reported as shown below. This paper need to clarified their pros and cons.

  • Continuous polymerase chain reaction microfluidics integrated with a gold-capped nanoslit sensing chip for Epstein-Barr virus detection https://doi.org/10.1016/j.bios.2021.113672
  • All-fiber all-optical quantitative polymerase chain reaction (qPCR), https://doi.org/10.1016/j.snb.2020.128681
  • A self-digitization chip integrated with hydration layer for low-cost and robust digital PCR. https://doi.org/10.1016/j.aca.2018.12.029
  • Absolute quantification and analysis of extracellular vesicle lncRNAs from the peripheral blood of patients with lung cancer based on multi-colour fluorescence chip-based digital PCR. https://doi.org/10.1016/j.bios.2019.111523
  • Multiplex amplification of target genes of periodontal pathogens in continuous flow PCR microfluidic chip. DOI: 10.1039/D1LC00457C
  • A Thermocycler Using a Chip Resistor Heater and a Glass Microchip for a Portable and Rapid Microchip-Based PCR Device. https://doi.org/10.3390/mi13020339

  1. For this fluorescent microscope, the authors set up a illuminating light in front of camera, even with an emission filter in between. However, the background noise is still very high. To solve the problem, the authors extracted RGB color to improve the contrast, with drawback of reducing sensitivity. Several efforts have developed portable darkfield fluorescent microscope as listed below. The authors need to compare this current system to previous efforts in the section of Results and Discussion.

  • Optofluidic Fluorescent Imaging Cytometry on a Cell Phone. https://doi.org/10.1021/ac201587a
  • Rapid imaging, detection and quantification of Giardia lamblia cysts using mobile-phone based fluorescent microscopy and machine learning. DOI: 10.1039/C4LC01358A
  • Colour compound lenses for a portable fluorescence microscope. https://doi.org/10.1038/s41377-019-0187-1
  • A Smartphone-based Diffusometric Immunoassay for Detecting C-Reactive Protein. https://doi.org/10.1038/s41598-019-52285-4

  1. The authors demonstrated the real-time PCR in Figure 5. However, the detection limit of this system was not showed. The authors need to carry out some extra experiments using different initial concentrations of nucleic acid to demonstrate the detection limit.

Round 2

Reviewer 2 Report

I am satisfied with the current version.